# Molecular Interactions of the Copper Chaperone Atx1 of *Paracoccidioides brasiliensis* with Fungal Proteins Suggest a Crosstalk between Iron and Copper Homeostasis

**DOI:** 10.3390/microorganisms11020248

**Published:** 2023-01-18

**Authors:** Marcos Antonio Batista de Carvalho Júnior, Lana O’Hara Souza Silva, Laurine Lacerda Pigosso, Aparecido Ferreira de Souza, Danize Eukales Menezes Lugo, Dayane Moraes, Kleber Santiago Freitas e Silva, Maristela Pereira, Célia Maria de Almeida Soares

**Affiliations:** Laboratorio de Biologia Molecular, Instituto de Ciências Biológicas, Universidade Federal de Goiás, Goiânia 70690-900, GO, Brazil

**Keywords:** *Paracoccidioides brasiliensis*, Atx1, protein–protein interaction, pull-down, mass spectrometry, Cyb5

## Abstract

*Paracoccidioides* spp. are endemic fungi from Latin America that cause Paracoccidioidomycosis, a systemic disease. These fungi present systems for high-affinity metal uptake, storage, and mobilization, which counteract host nutritional immunity and mitigate the toxic effects of metals. Regarding Cu mobilization, the metallochaperone Atx1 is regulated according to Cu bioavailability in *Paracoccidioides* spp., contributing to metal homeostasis. However, additional information in the literature on *Pb*Atx1 is scarce. Therefore, in the present work, we aimed to study the *Pb*Atx1 protein–protein interaction networks. Heterologous expressed *Pb*Atx1 was used in a pull-down assay with *Paracoccidioides brasiliensis* cytoplasmic extract. Nineteen proteins that interacted with *Pb*Atx1 were identified by HPLC-MS^E^. Among them, a relevant finding was a Cytochrome *b*_5_ (*Pb*Cyb5), regulated by Fe bioavailability in *Aspergillus fumigatus* and highly secreted by *P. brasiliensis* in Fe deprivation. We validated the interaction between *Pb*Atx1-*Pb*Cyb5 through molecular modeling and far-Western analyses. It is known that there is a relationship between Fe homeostasis and Cu homeostasis in organisms. In this sense, would *Pb*Atx1-*Pb*Cyb5 interaction be a new metal-sensor system? Would it be supported by the presence/absence of metals? We intend to answer those questions in future works to contribute to the understanding of the strategies employed by *Paracoccidioides* spp. to overcome host defenses.

## 1. Introduction

The host–pathogen interaction is supported by several molecular events, and pathogens’ infective success lies in their ability to overcome the conditions imposed by the host [1]. Among these events, the homeostasis of metals such as Iron (Fe), Zinc (Zn), and Copper (Cu) have been the subject of studies, since the control of cellular availability of these metals affects several attributes such as metabolism, stress response, and virulence factors [2,3]. Copper is a transition metal that is biologically present in the forms of Cu^+^ (cuprous) and Cu^2+^ (cupric). The ease of being used as an electron donor and acceptor easily allows the metal to be used as a cofactor for several enzymes, which makes copper indispensable for the vast majority of organisms [4]. This flexibility of movement between oxidation states also makes copper a metal with toxic potential, as it can damage biomolecules by generating reactive oxygen species. Therefore, strict metal control is needed to ensure homeostasis [5].

In fungi, regarding the machinery responding to cellular copper levels, the Atx1 chaperone is classically described as having a pivotal role in the copper buffering and trafficking. The roles played by Atx1 are well described in *Saccharomyces cerevisiae*, where the protein is known to protect against reactive oxygen species (ROS) in superoxide dismutase (SOD) non-expressing cells [6]. In the Baker’s yeast, Atx1 was still characterized for mobilizing copper from the high-affinity transporter (Ctr1) in the plasma membrane to the copper translocator ATPase (Ccc2), constituting a secretory pathway that is responsible for metalate, the multi-copper ferroxidase Fet3 that participates in iron uptake [7]. *S. cerevisiae* Atx1 also seems to be involved in the cell cycle, since its deletion produced transcriptional responses in genes related to this process [8]. The Atx1–Ccc2 axis in the secretory pathway is also used by *Cryptococcus neoformans* to metalate laccase, which culminates in melanin production [9]. In *Aspergillus fumigatus*, AtxA is thought to be responsible for supplying iron with high-affinity reductive iron uptake, when the siderophore biosynthesis pathway is impaired [10].

*Paracoccidioides* spp. are thermal dimorphic pathogenic fungi that cause paracoccidioidomycosis (PCM), an endemic disease of Latin America with remarkable importance in Brazil [11]. Once the environmental mycelia and conidia of the *Paracoccidioides* spp. fungi are inhaled, they start to be constantly challenged by the host’s innate immune defense mechanisms. In the lungs, these pathogens face alveolar phagocytes where they can survive and even multiply within the phagosomes, since *Paracoccidioides* spp. is a facultative intracellular pathogen [12]. The copper burst strategy, as a mechanism used by phagocytes to cause toxicity in intracellular pathogens, is well established [13,14]. However, we have previously demonstrated that when phagocytosed, *Paracoccidioides* spp. faces Cu deprivation [15]. It is important to note that *Paracoccidioides* is able to respond to both Cu deprivation and overload conditions produced in vitro [15,16].

*Paracoccidioides* spp. has genes homologous to those of *S. cerevisiae* that encode Cu uptake and homeostasis systems, including the metallochaperone Atx1 [3]. It has also been shown that copper overload causes global changes in *Paracoccidioides* cells, promoting metabolic adaptation, in addition to the accumulation of ergosterol and melanin [16]. However, *Paracoccidioides* cells subjected to Cu deprivation show increased beta oxidation, upregulation of Cu-independent detoxification systems, and increased Atx1 expression [15]. Although Atx1 is a central element in Cu metabolism, little is known about Atx1 from *Paracoccidioides brasiliensis*. In this sense, the study of its protein–protein interactions (PPI) can be useful, as these events are fundamental to many biological processes [17]. Therefore, in this work, we sought to contribute to the knowledge regarding the *Pb*Atx1 interactome, discovering new processes in which this protein is engaged.

## 2. Materials and Method

### 2.1. Strain Used and Culture Media

*P. brasiliensis* (*Pb*18-ATCC 32069) was employed in this study. The yeast cells were cultivated in semi-solid Fava Netto medium supplemented with glucose 4% (*w/v*) at a temperature of 36 °C [18]. The cells were grown at 36 °C for 48 h in liquid BHI medium containing 4% (*w/v*) glucose in order to perform the experiments.

### 2.2. Obtaining Protein Extracts

The yeast cultures were prepared by inoculating 250 mL of Fava Netto’s liquid medium with 10^6^ cells/mL, incubated under agitation at 36 °C for 72 h. The cells were centrifuged at 10,000× *g* for 5 min and to the pellet extraction buffer (20 mM Tris–HCl pH 8.8; 2 mM CaCl_2_) was added. To the suspension glass, beads were added, followed by processing on ice in the bead beater (BioSpec, Bartlesville, OK, USA) apparatus for 5 cycles of 30 s. The material was centrifuged at 10,000× *g* for 15 min at 4 °C. After collecting the supernatant, protein concentrations were determined using the Bradford reagent (Sigma Aldrich, Co., St. Louis, MO, USA); bovine serum albumin (BSA) was used as a standard.

### 2.3. Pull-Down Assays

Pull down assay was performed according to SILVA et al. (2019). Briefly, recombinant *Pb*Atx1 was immobilized in a Ni-NTA resin. A total of 300 μg of *P. brasiliensis* (*Pb*18) protein extract was incubated for 2 h at 4 °C under gentle agitation. Later, the column carrying bait and prey proteins was washed with native wash buffer (50 mM sodium phosphate buffer, 500 mM sodium chloride, 20 mM imidazole, pH 8.0) 5 times to diminish unspecific interactions or contaminants and then, the complex bait–preys was eluted with native elution buffer (50 mM sodium phosphate buffer, 500 mM sodium chloride, 250 mM imidazole, pH 8.0). The eluted sample was tryptic digested, and the peptides were separated via NanoUPLC-MSE and analyzed using the nanoACQUITY system (Waters Corporation, Milford, MA, USA) to identify proteins that interacted with *Pb*Atx1. As a control, 300 μg of *P. brasiliensis* (*Pb*18) protein extract was incubated with Ni-NTA without the immobilized *Pb*Atx1, and results were used as the negative control.

### 2.4. Sample Preparation for nanoUPLC-MS^E^

Proteins from the assay above were quantified using the Bradford reagent. Afterwards, 150 µg of proteins from three biological replica were individually prepared to perform high resolution liquid chromatography, on a nanoscale, coupled to mass spectrometry with independent data acquisition (nanoUPLC-MS^E^) [19]. Initially, 10 µL of 50 mM NH_4_HCO_3_, pH 8.5, were added to the samples. Next, 75 µL of a 0.2% (*w/v*) RapiGEST^TM^ solution (Waters, Milford, MA, USA), a surfactant, was added; the mixture was incubated at 80 °C for 15 min. Later, 2.5 µL of 100 mM DTT, a disulfide bridge reducing agent, was added and samples were incubated for 30 min at 60 °C; samples remained there until they reached room temperature. Next, 2.5 µL of 300 mM Iodoacetamide, an alkylating agent, was added, followed by incubation for 30 min at room temperature, in absence of light. For tryptic digestion, 30 µL of a 0.05 µg/µL trypsin solution (Promega, Madison, WI, USA) was added and samples were incubated at 37 °C for 16 h. Then, 30 µL of 5% trifluoroacetic acid (*v/v*) were added for precipitation of the surfactant, with incubation at 37 °C for 90 min. After centrifugation at 13,000× *g* for 30 min, at 4 °C, samples were concentrated in vacuo. Peptides were resuspended in 80 µL of a solution containing 20 mM ammonium formate, pH 10, and 200 fmol/µL of PHB (Rabbit Phosphorylase B) (Waters Corporation, Manchester, UK) (MassPREP^TM^ protein). PHB was used as an internal standard for the subsequent quantification of peptides.

### 2.5. High Performance Liquid Chromatography at Nanoscale Coupled to Mass Spectrometry

The digested samples underwent high resolution liquid chromatography, on a nanoscale, in the ACQUITY UPLC^®^ M-Class system (Waters Corporation, Milford, MA, USA). Peptide fractionation utilized a reverse phase pre-column UPLC M-Class Peptide 5 µm BEH C18 130 Å 300 µm × 50 mm (Waters Corporation, Milford, MA, USA), in a flow of 0.5 μL/min with an initial condition of acetonitrile (ACN) of 3%; this was the first dimension. Peptides were subjected to 5 fractionations (F1–F5), at a linear gradient of ACN concentrations (F1-11.4%; F2-14.7%; F3-17.4%; F4-20, 7% and F5-50%). For the second dimension, each fraction was eluted in a trapping column, 2D Symmetry^®^ 5 µm BEH100 C18, 180 µm × 20 mm (Waters Corporation, Milford, MA, USA), and introduced through an analytical column separation Peptide CSH™ BEH130 C18 1.7 µm, 100 µm × 100 mm (Waters, Milford, MA, USA), in a flow of 0.4 μL/min at 40 °C. After, 200 fmol/µL of the human (Glu1) -Fibronopeptide B protein (GFP—Sigma-Aldrich, St. Louis, MO, USA) was used for mass calibration, which was measured every 30 s and in a constant flow of 0.5 µL/min. The peptides were identified and quantified by a Synapt G1 MS^TM^ mass spectrometer (Waters, Milford, MA, USA) equipped with a NanoElectronSpray source and two mass analyzers: a quadrupole and a time-of-flight (TOF) operating in V-mode. Data were obtained using the instrument in the MSE mode, which switches the low energy (6 V) and elevated energy (40 V) acquisition modes every 0.4 s. For the biological replicates, each condition was analyzed through 3 experimental replicates.

### 2.6. Spectra Processing and Proteomic Analysis

Data processing used ProteinLynx Global Server version 3.0.2 (PLGS) software (Waters Corporation, Milford, MA, USA), which allowed the determination of the exact mass retention time (EMRT) of the peptides, as well as how to infer their molecular weight through the mass/charge ratio (*m/z*). For peptide identification, the spectra obtained (together with reverse sequences) were compared with sequences of *P. brasiliensis* proteome from Uniprot database (https://www.uniprot.org/uniprot/?query=paracoccidioides+brasiliensis+strain+pb18&sort=score, accessed on 6 February 2020). For protein identification, the following criteria were considered: detection of a minimum of 2 ions per fragment of peptides; 5 by protein fragments; determination of a minimum of 1 peptide per protein; 4% at most detection rate of false positive; cysteine carbamidomethylation; methionine oxidation; serine, threonine and tyrosine phosphorylation; a trypsin lost cleavage site was allowed. Expression Algorithm (ExpressionE), which is part of the PLGS software [20], was used for the analysis of differential expression. Proteins identified exclusively in the interaction with *Pb*Atx1 were functionally categorized according to UniProt (https://www.uniprot.org/, accessed on 1 June 2022) and KEGG database (http://www.genome.jp/kegg/, accessed on 1 June 2022). Sequence annotation was assessed using a BlastP algorithm (http://blast.ncbi.nlm.-nih.gov/Blast.cgi, accessed on 1 June 2022). All on-line algorithms were used in default parameters.

### 2.7. Recombinant Proteins and Polyclonal Antibodies

These were previously obtained by Petito et al. (2020) [15] and Souza et al. (2022) [21], respectively, in our group.

### 2.8. SDS-PAGE and Western Blotting 

*P. brasiliensis* extract or pull-down elution (30 µg for both) were separated by polyacrylamide gel electrophoresis (12% SDS-PAGE) and transferred to a nitrocellulose membrane that was then incubated with polyclonal primary antibodies anti-*Pb*Cyb5, at a 1:250 dilution, for 2h at room temperature. After washing with buffer (washing buffer: PBS 1X, 0.1% (*v/v*) Tween 20), membranes were incubated with peroxidase-coupled anti-mouse IgG secondary antibody (1:1000 dilution). The negative control was obtained with pre-immune mouse serum (1:250 dilution). The reaction was revealed with ECL Western Blotting Analysis System (GE Healthcare, Chicago, IL, USA) or DAB (3,3′-diaminobenzide tetrahydrochloride, Sigma Aldrich, Co., St. Louis, MO, USA) in a chemiluminescent imager (Amersham Imager 600, GE Healthcare, Chicago, IL, USA). Negative controls were obtained with mouse pre-immune serum.

### 2.9. STRING Database Analysis

The STRING (Search Tool for the Retrieval of Interacting Genes/Proteins) database is based on the prediction and integration of PPI data. It considers direct and indirect associations to predict possible interactions, even for organisms where experiments are unavailable when homology is identified. The accession number of Atx1 was used to identify possible interactions performed by this protein, using STRING database version 11.5 [22]. In the analysis, we applied the highest confident score (0.900) and considered the active interaction sources such as text mining, experiments, databases, and co-expression.

### 2.10. Prediction of PbAtx1 and PbCyb5 Structures 

The I-TASSER (Iterative Threading Assembly Refinement) server [23] modelled the three-dimensional structures of *Pb*Atx1 (Copper metallochaperone Atx1—PADG_02352) and *Pb*Cyb5 (Cytochrome b5—PADG_03559) from *P. brasiliensis*. This server predicts protein structures based on templates of homologous proteins experimentally determined and available on the PDB (protein data bank). I-TASSER relies on fold recognition and Monte Carlo simulations in order to score homologous fragments [24]. 

Generally, the secondary structure of the target protein is predicted by PSSpred (Protein Secondary Structure Prediction) and threading templates are recognized by LOMETS (Local Meta-Threading-Server) [25]. Resulting fragments are classified into clusters according to energy levels and conformational states through SPICKER [26]. The latter optimize the predicted conformation, according to native similar structures. Eventually, the predicted structure is subjected to molecular dynamics optimization and prediction of its biological function by COACH [27]. The quality of the Atx1 three-dimensional structure was evaluated by the MolProbity server [28].

The model obtained was also compared with the already resolved crystallographic structure of Atx1p from *S. cerevisiae* (PDB Entry ID: 1FD8). The model was also submitted to the ConSurf server, which reports on the evolutionary conservation degree of amino acids along the structure of a protein [29].

### 2.11. Molecular Docking

The ClusPro protein–protein docking server [30] was used to establish the best protein complex conformations between *Pb*Atx1 and *Pb*Cyb5. The model with the lowest score and consequently with the highest balanced conformation was selected. Then, the KFC2 server [31] was useful in order to recognize residues within the interaction interface, including hotspots. The CoCoMAPS server was used to analyze the interaction interface of the complex through intermolecular contact maps. This server aided the calculation of hydrogen bond distances between the interacting residues [32]. Lastly, the PyMOL (The PyMOL Molecular Graphics System, Version 1.2r3pre, Schrödinger, New York, NY, USA) molecular visualization program was employed to visualize the best conformations of the binding proteins and the residue’s interaction within the interface of interaction.

### 2.12. Far-Western Blot Analyses

The protocol of far-Western was previously described [33,34]. In total, 20 µg of the recombinant *Pb*Cyb5 protein (PADG_03559) was electrophoretically transferred to a Hybond nitrocellulose membrane (GE Healthcare, Piscataway, NJ, USA) and incubated for 1 h at room temperature with the recombinant purified *Pb*Atx1 protein, diluted at 35 µg/mL in blocking buffer (PBS 1X, 5% (*w/v*) non-fat milk and 0.1% (*v/v*) Tween 20). The membrane was washed and incubated with anti-*Pb*Atx1 polyclonal antibodies, diluted 1:250 in blocking buffer. After washing, the membranes were incubated with peroxidase-coupled anti-mouse IgG secondary antibody (1:1000 dilution). As a negative control, the same treatment described above was carried out with BSA transferred to the Hybond nitrocellulose membrane. The reaction was revealed with ECL Western Blotting Analysis System (GE Healthcare, Chicago, IL, USA) or DAB (3,3′-diaminobenzide tetrahydrochloride, Sigma Aldrich, Co., St. Louis, MO, USA) in a chemiluminescent imager (Amersham Imager 600, GE Healthcare, Chicago, IL, USA). 

## 3. Results

### 3.1. Overview of PbAtx1 Protein–Protein Interactions

To search for *P. brasiliensis* proteins that might bind to *Pb*Atx1, we performed a pull-down assay followed by mass spectrometry. For that, recombinant *Pb*Atx1 was immobilized on Ni-NTA resin, and incubated with *P. brasiliensis* proteins extract. Next, the eluate of this assay was submitted to UPLC-MS^E^ to identify proteins that might interact with *Pb*Atx1. This analysis identified 19 proteins, not present in the control assay. As depicted in Table 1, the minimum of peptides for each protein was three, and ranged to a maximum of sixteen peptides, demonstrating the accuracy of the identification, also corroborated by the obtained score.

### 3.2. Pull-Down Validation by Western Blot Analysis

To validate the pull-down assay, we performed Western blot analysis with the eluted proteins using the polyclonal antibodies anti-*Pb*Cyb5, produced upon the immunization of BALB/c mice with the recombinant *Pb*Cyb5 protein [21] (Figure 1). Anti-*Pb*Cyb5 reacted with the pull-down sample, thus corroborating the assay depicted in Table 1, confirming that this protein is among *Pb*Atx1 partners.

### 3.3. In Silico Seek for PbAtx1 Interaction Network

We also sought to analyze, through an in silico approach, which proteins are predicted to interact with *Pb*Atx1 by using the Search Tool for the Retrieval of Interacting Genes/Proteins (STRING v11.5). The analysis revealed eight proteins that might interact with *Pb*Atx1 (Figure 2). Among them, we highlight the presence of *Pb*Cyb5 which was also identified in the pull-down assay, reinforcing that this protein might be a *Pb*Atx1 interaction partner. Therefore, we selected this protein for further analysis.

### 3.4. Molecular Modeling of PbAtx1 and PbCyb5 Resulted in Stable and Conserved Conformational Structures

Due to the interaction of Atx1 and Cytochrome b5, we performed molecular modeling of *Pb*Atx1 and *Pb*Cyb5 in order to corroborate experimental data. The *Pb*Atx1 contains a heavy-metal-associated domain signature, identified through analysis using the InterPro server [35]. The alignment of *Pb*Atx1 primary structure with fungal homologues is depicted in Appendix A. The heavy-metal-associated domain has two conserved cysteines that contribute to the binding of Atx1 to metal atoms, and the transfer of this metal to a receptor protein (Appendix A).

The domain structure is well-defined and is composed of a four-chain antiparallel beta sheet and two sandwich-shaped alpha helices (Figure 3A). The three-dimensional structure of *Pb*Atx1, according to the molecular modeling (Figure 3A), resulted in reliable scores (C-score = 0.06, TM-score = 0.72 ± 0.11 and RMSD = 3.4 ± 2.4 Å). This ensures that the predicted model is close to in vivo conditions. The model of the *Pb*Cyb5 protein resulted in scores within the expected range for conserved proteins with a stable conformational structure (C-score = −1.60, TM-score = 0.52 ± 0.15 and RMSD = 8.0 ± 4.4 Å). The *Pb*Cyb5 protein comprises antiparallel beta sheets, alpha helices, and a loose loop region (Figure 3B).

### 3.5. PbAtx1 Interacts with PbCyb5

We selected the *Pb*Cyb5 protein to further investigate its interaction with *Pb*Atx1. We selected this protein firstly because Cyb5 is was obtained in the pull down assays, and secondly for the evidence of its interaction with Atx1 in *S. cerevisiae* [36]. To date, this is the first time these interactions have been described in *P. brasiliensis*, and the first study to molecularly identify the interaction interface between *Pb*Atx1 and *Pb*Cyb5. The interaction model of the binary complex formed between *Pb*Atx1 and *Pb*Cyb5 corresponds to the model of the lowest free energy level and is represented in Figure 4.

The interaction between *Pb*Atx1 and *Pb*Cyb5 is favored by the presence of three main hotspots. In addition, two hotspots’ residues are on the surface of the interaction interface, and the other one is buried within this region (Figure 4). They interact via hydrogen bonds to keep the complex energetically stable. The residues E56 (glutamic acid) and E59 establish two hydrogen bonds each with residues K28 (lysine) and R25 (arginine) of Atx1, respectively. These interactions are in the order of 2 Å (Figure 4A). The residue I115 (isoleucine), which is one of the main hotspots, establishes two hydrogen bonds, one with residue A22 (alanine) with a distance of 3.5 Å and with R25, with a distance of 3.4 Å (Figure 4A).

Figure 4B shows a denser region of hotspots at the interaction interface between *Pb*Atx1 and *Pb*Cyb5. These hotspots are close to each other and inserted in a region of the interface where they establish interactions with distances of the order of 3.0 Å (Figure 4B). The residue hotspot K67 interacts with T95 (threonine) at a distance of 3.3 Å and the hotspot F108 (phenylalanine) interacts with T68 at 3.5 Å, S97 (serine) at 3.0 Å, and T68 at 3.4 Å.

Then, to certify this interaction, we realized a far-Western blot with recombinant *Pb*Cyb5 bound to a nitrocellulose membrane and incubated with recombinant *Pb*Atx1. The reaction was revealed after incubation with anti-*Pb*Atx1 antibodies and anti-mouse IgG coupled to peroxidase. Corroborating with the pull-down and Western blot assays, *Pb*Cyb5 showed interaction with *Pb*Atx1 (Figure 4C), confirming that these proteins interact with each other.

## 4. Discussion

In this work, we continued our studies on metal homeostasis in the genus *Paracoccidioides.* The protein Atx1 was previously identified in *Paracoccidioides* spp. by proteomic analysis of yeast cells in copper deprivation. To better understand the role of Atx1 in the genus, we studied the molecular interactions of the protein by using experimental and in silico approaches. Among the *Pb*Atx1 partner proteins identified through pull-down assay (Table 1), we found argininosuccinate synthase (AS—PADG_00888). This protein participates in the urea cycle, catalyzing the condensation of citrulline and aspartate, resulting in L-argininosuccinate [37]. In *P. brasiliensis* and other fungal species such *A. fumigatus* and *S. cerevisiae*, an additional role, other than a copper chaperone, has been proposed for AS in the siderophore biosynthesis, once the downstream steps in the urea cycle culminate in the formation of ornithine, a precursor of siderophores [34,38]. Thus, since Atx1 is also reported to participate in iron acquisition processes through the metalation of ferroxidases, as mentioned before, it is not unlikely that more crosstalk events between copper and iron homeostasis pathways could occur, such as an interaction between *Pb*Atx1 and *Pb*AS as described here.

The transition between the two different oxidation states of copper contributes distinctly to the way this metal binds to proteins. The Cu^+^ ion, for example, has a higher affinity for thiol groups on cysteine and methionine residues. On the other hand, the oxidized form Cu^2+^ binds preferentially to the oxygen of residues such as glutamate and aspartate, or to the imidazole present in histidine. [39]. Cu^2+^ binds to Atx1p in *S. cerevisiae* through the conserved metal-binding motif *MXCXXC* (where, M = Methionine, C = Cysteine, and X = variable amino acids) [40]. Through in silico analyses, we verified that *Pb*Atx1 retained this copper-binding motif (Appendix A). Additionally, pull-down analysis coupled to mass spectrometry revealed that *Pb*Atx1 interacts with both thioredoxin (TRX—PADG_01551) and thioredoxin reductase (TRX-R—PADG_03161) (Table 1). These two proteins are known to act as a system that reduces disulfide bonds in cysteine residues present in oxidized proteins [41]. Interestingly, copper binding to the human Atox1 is regulated through the catalytic reduction of the disulfide bonds between cysteine residues, mediated by a protein of the thioredoxin superfamily [42]. Additionally, in the bacteria *Bradyrhizobium japonicum*, the thioredoxin (TlpA) was also reported to be responsible for catalyzing the reduction of disulfide bonds in a copper chaperone (ScoI) and in the copper centers of cytochrome oxidase (CoxB), thus providing adequate metalation of these proteins [43]. Thus, we believe in a possible regulatory role of copper binding to the *MXCXXC* motif in *Pb*Atx1.

We were able to demonstrate, by pull down and far-Western analysis, that *Pb*Atx1 interacts with *Pb*Cyb5 (Table 1 and Figure 4). The literature shows that Cyb5 is a multifunctional protein, according to studies in *A. fumigatus* (CybE/Cyb5). Cytochrome b5 acts as an electron donor in the ergosterol biosynthetic pathway. Genetic engineering to disrupt Cyb5 function affected membrane homeostasis [44]. Cyb5 also plays an essential role in the process of resistance to azoles, in addition to being regulated by the bioavailability of Fe [45]. Both studies point to Cyb5 as a good target for drugs in *A. fumigatus*, given the relevance of the processes promoted by the protein. A recent study of the exoproteome of *P. brasiliensis* demonstrated that Fe deprivation promotes the regulation of several proteins, including *Pb*Cyb5. In vitro interaction assays demonstrated that *Pb*Cyb5 is capable of binding to inorganic Fe, and this finding was supported by molecular dynamics analyses [21].

The structural models predicted for the *Pb*Atx1 and *Pb*Cyb5 structures by homology correspond to the lowest energy level, reflecting a more stable conformation for the protein complexes, and a more reliable representation of their native form [46]. Certain factors contribute to the interaction of *Pb*Atx1 with proteins related to copper metabolism, electron transport, and detoxification proteins. The motifs presented by the *Pb*Atx1 structure are anchoring regions for metalloproteins, for example. The heavy metal-associated domain, present in *Pb*Atx1, is conserved and has approximately 30 amino acid residues, and the binding stoichiometry is one copper ion per binding domain (Figure 3A). This domain is also present in proteins with redox functions that transport or detoxify heavy metals, for example, ATPases, electron transporters, and copper chaperones [47]. These classes of proteins were identified in the experimental assays of the present research.

The study of the interaction interface between *Pb*Atx1 and *Pb*Cyb5 identified hotspots’ residues. These are residues within the protein–protein interface that contribute to a reduced free binding energy and consequently, a stable conformation of the complex [48]. Generally, hotspots are gathered in regions of the interface (Figure 4B) [49], and residues neighboring them can also establish hydrogen bonds to decrease the free energy of the complex and assist in building the most stable conformation (Figure 4). Despite the evidence of interaction between Atx1 and Cyb5, to date, this is the first time these interactions have been described in *P. brasiliensis*.

It is known that Atx1 delivers Cu to Ccc2 in the vacuole. Multicopper oxidases then gain access to the metal. These proteins act in the reductive Fe uptake pathway [50]. Would this interplay between Cu bioavailability and Fe homeostasis be the justification for the *Pb*Atx1–*Pb*Cyb5 interaction? Would these proteins form a Fe bioavailability sensor system for the activation of high affinity systems for Fe uptake? Would the *Pb*Atx1–*Pb*Cyb5 interaction be supported by the presence/absence of Fe or Cu? We are aware that all these issues need to be investigated to elucidate the process.

## Figures and Tables

**Figure 1 microorganisms-11-00248-f001:**
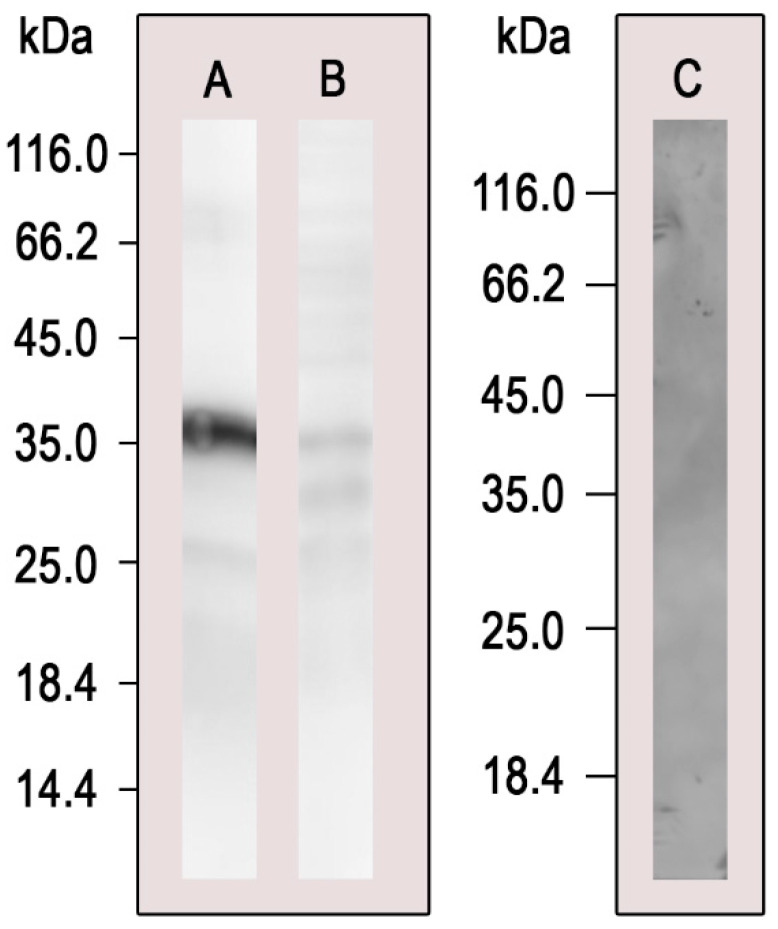
Reactivity of anti-*Pb*Cyb5 polyclonal antibodies with proteins eluted from the pull-down assay. (**A**)—Pull-down eluted proteins were submitted to SDS-PAGE, transferred to a nitrocellulose membrane, and incubated with anti-*Pb*Cyb5 polyclonal antibodies (dilution 1:250). This positive reaction with anti-*Pb*Cyb5 in pull-down elution corroborates with our mass spectrometry data indicating *Pb*Cyb5 as a *Pb*Atx1 partner protein. (**B**)—As a positive control to test the reactivity of the antibody, cytoplasmatic protein extracts obtained from *P. brasiliensis* were also incubated with anti-*Pb*Cyb5 (dilution 1:250). (**C**)—Negative control using pre-immune serum (1:1000).

**Figure 2 microorganisms-11-00248-f002:**
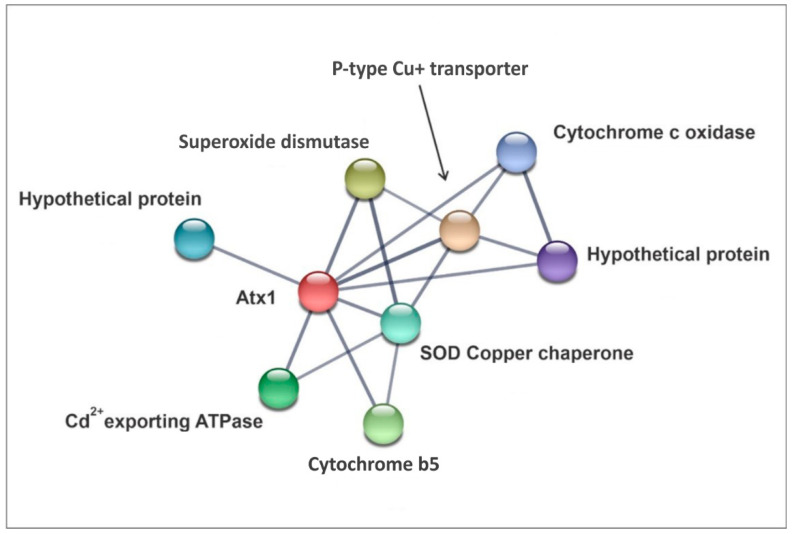
PbAtx1 interaction network. STRING prediction of *P. brasiliensis* Atx1 (PADG_02352) protein–protein interactions. Eight proteins were predicted to interact with *Pb*Atx: P-type Cu+ transporter (PADG_01582), Cytochrome c oxidase (PADG_07058), Hypothetical protein (PADG_04769); Copper Chaperone for Superoxide dismutase (PADG_01400), Cytochrome b5 (PADG_03559), Cd^2+^-exporting ATPase (PADG_03376), Superoxide dismutase (Cu-Zn) (PADG_07418), and Hypothetical protein (PADG_01124).

**Figure 3 microorganisms-11-00248-f003:**
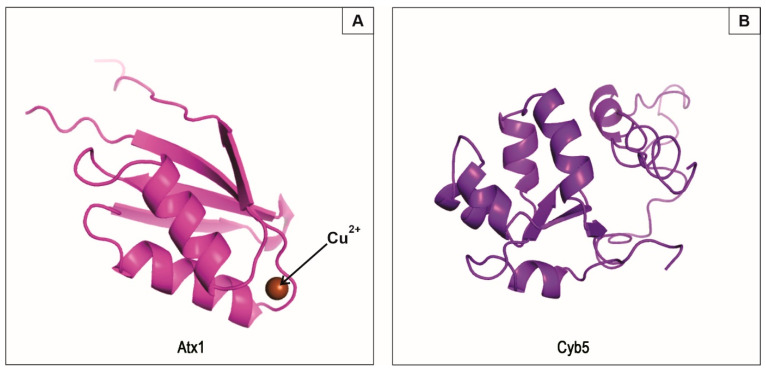
Molecular modeling of the *P. brasiliensis* proteins *Pb*Atx1 and *Pb*Cyb5. (**A**)—Conformational structure of modeled *Pb*Atx1 showing the copper ion disposition. This conserved domain contains cysteine residues and contributes to the interaction of the protein with the copper ion. (**B**)—Predicted structure for the *Pb*Cyb5 protein, showing in its secondary structure regions of alpha helix, beta sheet, and a terminal region rich in flexible loops.

**Figure 4 microorganisms-11-00248-f004:**
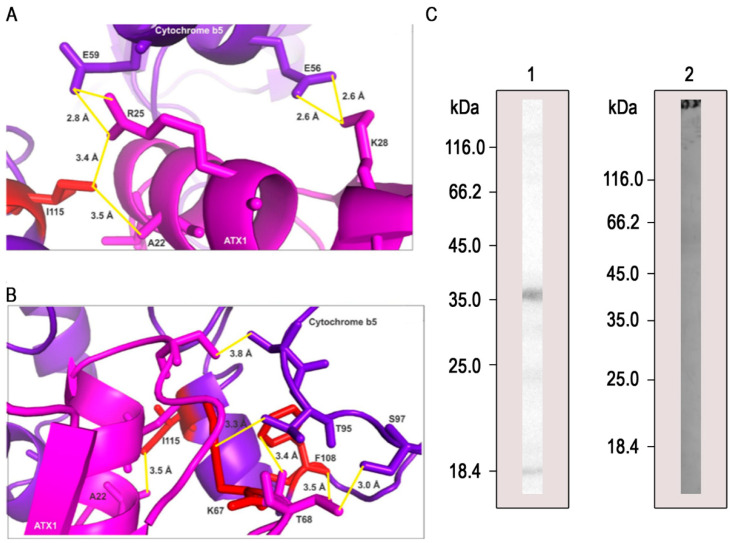
Interaction of *Pb*Atx1 and *Pb*Cyb5. (**A**)—The hotspot residue I115 forms hydrogen bonds with A22 and R25, and neighboring residues E56 and E59 form hydrogen bonds with K28 and R25, respectively. (**B**)—Residues K67 and F108 were also classified as hotspots and interact with residues present within the interface. This region contributes most, energetically, to the conformation of the complex once there are more hotspot residues and they are close to each other. Residues in red correspond to hotspots. (**C**)—(1) Far-Western of 20 µg of *Pb*Cyb5 fractionated by SDS-PAGE, transferred to a nitrocellulose membrane, and incubated with the purified *Pb*Atx1 (35 μg/mL). (2) Negative control of BSA (60 µg) also transferred to a nitrocellulose membrane and incubated with purified *Pb*Atx1 (35 μg/mL). The interaction was detected with anti-*Pb*Atx1 polyclonal antibodies (1:500).

**Table 1 microorganisms-11-00248-t001:** *Pb*ATX—interacting proteins by using pull-down assays identified through LC-MS/MS.

Accession ^a^	Description	Score	Sequence Coverage	Peptides ^b^
**PADG_00888**	Argininosuccinate synthase	797.87	14.22	6
**PADG_01886**	Adenosylhomocysteinase	1375.59	27.52	14
**PADG_04167**	Aspartyl aminopeptidase	863.77	21.96	13
**PADG_03449**	Isopentenyl-diphosphate delta-isomerase	872.73	13.28	8
**PADG_00221**	Short-chain dehydrogenase	1357.70	21.4	9
**PADG_05081**	Aldehyde dehydrogenase	1362.79	30.8	16
**PADG_00206**	Mitochondrial zinc maintenance protein 1, mitochondrial	2151.42	10.48	3
**PADG_01551**	Thioredoxin reductase	1759.66	22.35	8
**PADG_03161**	Thioredoxin	906.63	33.73	13
**PADG_02181**	HAD-superfamily hydrolase	718.42	24.36	9
**PADG_00128**	Tubulin alpha-2 chain	655.55	21.29	9
**PADG_07422**	Serine proteinase	1204.78	5.66	4
**PADG_00688**	F-type H+-transporting ATPase subunit H	1610.02	36.36	5
**PADG_03559**	Cytochrome-b5	955.84	35.51	3
**PADG_07883**	hypothetical protein	790.64	35.8	4
**PADG_04855**	hypothetical protein	810.68	20.31	10
**PADG_05110**	hypothetical protein	703.68	19.5	6
**PADG_00449**	hypothetical protein	1232.59	13.7	16
**PADG_03788**	hypothetical protein	811.28	15.03	8

^a^ Accession number according to FungiDB Database (http://fungidb.org/fungidb/, accessed on 1 June 2022). ^b^ Number of peptides related to each protein identified by LC-MS/MS.

## Data Availability

The data presented in this study are available on request from the corresponding author. The data are not publicly available because it is partially related to a subsequent publication.

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
