# Peer review of "Molecular Interactions of the Copper Chaperone Atx1 of Paracoccidioides brasiliensis with Fungal Proteins Suggest a Crosstalk between Iron and Copper Homeostasis"

_microorganisms, 2023, doi:10.3390/microorganisms11020248_

Round 1

Reviewer 1 Report

In this study, the authors investigated the interaction interface between PbAtx1 and PbCyb5.

One of the interesting finding that they discovered is that they found the evidence of interaction between Atx1 and Cyb5 in P. brasiliensis.

Author Response

According to reviewer 1, one of the interesting findings that they discovered is that they found the evidence of interaction between Atx1 and Cyb5 in P. brasiliensis. We thank the reviewer for the comment.

Reviewer 2 Report

The authors investigated the protein-protein interaction of the copper chaperone Atx1 in Paracoccidioides brasiliensis. The study starts promising with the pull-down assay to identify the interaction partners, but unfortunately, in the following the manuscript is very descriptive and the only novel finding appears to be the used organism, as all found interactions were already described for other fungi. The questions raised by the authors at the end of the discussion are very interesting, but in the current version, the manuscript appears to me as a mixture of preliminary experiments and a review about Atx1 interactions in fungi. I will point out my concerns in the following:

Major:

1.) The use of models for predictions is fine, but the findings have to be proven somehow by experimental data. I know that the genetic modification of P. brasiliensis is not feasible, but the importance of certain amino acids for the interaction could also be shown by purification of a recombinant or heterologous expressed modified cyb5 protein. The problem with models is also shown in the manuscript, as only one protein that was found in the pull-down assay was also predicted by the interaction network (Figure 2). Either the proof of the predicted models or the answer of at least one of the questions raised in the discussion would make the manuscript much more interesting to readers.

2.) I’m not convinced by the positive control in Figure 1. The band can hardly be seen, but if you consider the weak band at 35 kDa as a positive band, there are also several other weak bands. Can the authors explain, if this is due to unspecific binding of the antibody or different modifications of cyb5? Also in this regard, how much protein was loaded on the gel? It’s not written in the M&M section. Did the authors try to load higher protein amounts of the cytoplasmatic extract to gain stronger bands?

Minor:

There are some small typos in the manuscript, which will probably be corrected by proof reading. But please correct the cooper in the title to copper. This directly catches the eye.

If the authors compare the PbAtx1 with Atx1 in other fungi, it would maybe also be interesting to show the alignment of the conserved metal binding motif MXCXXC in Atx1 of different fungi.

The first 4 times “°C” is written in the M&M section, it’s written in 4 different ways. Please stick to one version.

Reviewer 3 Report

This paper describes studies to find new candidates for interaction with PbAtx1, the results are of interest and suggest new connections partners for Atx1 and a new connection to metal sensing. My major concern considers the physical analysis between Atx1 and CytB5 via far Western analysis and absence of correct controls. No details are provided on SDS-PAGE and proper controls are missing to show that correctly folded proteins interact

comments

-  mistake in the title : cooper

- line 57 thermodimorphic or, better, thermal dimorphic 

- line 173 names in not in full capitals

- line 176 no details on SDS-PAGE are provided. In Fig 1 reactivity of CytB5 serum was used to detect the protein. is this protein before SDS-PAGE denatured by SDS and heat? or is there native protein analyzed?

This question becomes critical in Fig 4 where the interaction via far-Western is studied between CytB5 that was first  separated by SDS-PAGE and Atx1. It should be described how the CytB5 protein was treated here and separated via electrophoresis. If this was regular SDS-PAGE and the CytB5 protein was denatured it is difficult to imagine how this interacts with Atx1 since the interaction area is very specific and required correctly folded protein as seen in Fig 4A. This suggests that Atx1 might interact on gel in an unspecific manner with some sequence. A control is required with denatured Atx1 here

If no native gelelectrophoresis of CytB5 is used it should actually be included here in far Western analysis of Fig 4. In this far Western study with native CytB5 separated and Atx1 for binding, an additional control is required. Atx1 should also used as denatured protein in the far Western to show that correctly folded Atx1 is only able to bind to native CytB1. If this does not occur it should be discussed since it compromises the conclusion of interactions between these 2 partners

- line 258 fig 1; the positive control shows many bands, the serum seems not very specific. The authors should comment on this. They should also include a lane with the recombinant protein CytB5 used in immunization to show specificity. 

Round 2

Reviewer 2 Report

The authors discussed and implemented all my concerns.

Reviewer 3 Report

The manuscript was correctly updated by the authors.